# Evaluating the Stress State and the Load-Bearing Fraction as Predicted by an In Vivo Parameter Identification Method for the Abdominal Aorta

**DOI:** 10.3390/medsci13010009

**Published:** 2025-01-27

**Authors:** Jerker Karlsson, Jan-Lucas Gade, Carl-Johan Thore, Carl-Johan Carlhäll, Jan Engvall, Jonas Stålhand

**Affiliations:** 1Unit of Cardiovascular Sciences, Department of Clinical Physiology in Linköping, Department of Medical and Health Sciences, Linköping University, SE-581 83 Linköping, Sweden; carl-johan.carlhall@liu.se (C.-J.C.); jan.engvall@liu.se (J.E.); 2Solid Mechanics, Department of Management and Engineering, Linköping University, SE-581 83 Linköping, Sweden; jan-lucas_gade@gmx.com (J.-L.G.); carl-johan.thore@liu.se (C.-J.T.); jonas.stalhand@liu.se (J.S.)

**Keywords:** abdominal aorta, in vivo, stress state, load-bearing fraction, in silico, evaluation

## Abstract

**Background:** Arterial mechanics are crucial to cardiovascular functionality. The pressure–strain elastic modulus often delineates mechanical properties. Emerging methods use non-linear continuum mechanics and non-convex minimization to identify tissue-specific parameters in vivo. Reliability of these methods, particularly their accuracy in representing the in vivo stress state, is a significant concern. This study aims to compare the predicted stress state and the collagen-attributed load-bearing fraction with the stress state from in silico experiments. **Methods:** Our team has evaluated an in vivo parameter identification method through in silico experiments involving finite element models and demonstrated good agreement with the parameters of a healthy abdominal aorta. **Results:** The findings suggest that the circumferential stress state is well represented for an abdominal aorta with a low transmural stress gradient. Larger discrepancies are observed in the axial direction. The agreement deteriorates in both directions with an increasing transmural stress gradient, attributed to the membrane model’s inability to capture transmural gradients. The collagen-attributed load-bearing fraction is well predicted, particularly in the circumferential direction. **Conclusions:** These findings underscore the importance of investigating both isotropic and anisotropic aspects of the vessel wall. This evaluation advances the parameter identification method towards clinical application as a potential tool for assessing arterial mechanics.

## 1. Introduction

Arterial mechanics play an important role in the normal functionality of the cardiovascular system [1,2] and impact many cardiovascular diseases during their initiation and development [3,4,5,6]. Arterial mechanics are often characterized in terms of the intrinsic material properties of the tissue, such as stiffness. Popular clinical measures of stiffness are the pressure–strain elastic modulus [7], the stiffness index [8], and pulse wave velocity [9]. Although widely used, these measures reflect the overall stiffness of the arterial wall and typically rely on linearized and homogenized material behavior. In recent years, several studies have addressed these shortcomings and proposed methods to identify patient-specific parameters for more realistic material models [10,11,12,13,14,15,16]. These studies use nonlinear continuum mechanical models in combination with nonlinear parameter identification and estimate material parameters by fitting the models to a clinically measurable response, typically the pressure–radius curve.

Although continuum mechanical models may potentially offer more details about the mechanics of the artery, they all encounter a fundamental challenge: the scope of clinical in vivo measurement is limited compared to traditional ex vivo experiments. This limitation inevitably restricts the number of parameters in the model that can be uniquely identified [13,15,17]. Several strategies can be employed to reduce the model’s complexity and obtain a unique set of identified parameters. For instance, physiologically motivated constraints can be introduced during parameter identification to narrow down the parameter space [12,14]. Alternatively, the arterial wall can be approximated as a homogeneous membrane [11,13,16].

A critical issue with patient-specific in vivo methods is the question of reliability related to the accuracy of the identified parameters. A first attempt to study this issue was made by creating a finite element (FE) model of an artery and use it as an in silico experiment to generate an in silico pressure–radius curve on which parameter identification was performed [12]. The identified parameters could then be directly compared with their counterparts in the FE model. There was only one subject in the study, however, and the results cannot be generalized. The same principle was used on a larger group of in silico (mock) experiments to validate the parameter identification method proposed in [13] for the human abdominal aorta (see Figure 1) [18]. The results showed good agreement in the parameters for physiologically relevant in silico arteries.

The reliability of patient specific in vivo methods cannot be assessed by analyzing the parameters alone; the mechanical state must also be considered. From a physiological point of view, the stress state is a crucial factor for tissue response, particularly in growth and remodeling [6,19,20,21,22]. This observation immediately raises three questions: First, to what extent will the discrepancy of the identified parameters impact the stress state? Second, the membrane assumption in [18] neglects transmural stress distribution, whereas the stress in a pressurized thick-walled cylinder is highest at the luminal side and drops off towards the outer boundary. This suggests that significant stress differences might be observed between the membrane model and the thick-walled model, particularly at high pressures. On the other hand, the residual stress redistributes the transmural stress and results in a more uniform stress field at physiological pressures [23,24]. Third, how are the predicted load-bearing fractions of elastin and collagen affected by the previous two points? In this paper, we aim to answer these questions by investigating the stress state predicted by the in vivo parameter identification method proposed in [18] by comparing it to in silico experiments of the abdominal aorta.

## 2. Material and Methods

### 2.1. Mechanical Models for Arteries

In this section, we first describe the general continuum model (a thick-walled model) for the in silico experiments, which is used to generate in silico pressure–radius data. Following this, the simplified constitutive membrane model (a thin-walled model) that is used for parameter identification is presented. This has been previously described and is briefly outlined below [13,18].

#### 2.1.1. General Continuum Model

##### Kinematics

The abdominal aorta is considered a residually stressed, homogeneous, and incompressible cylinder subjected to an external load in the form of internal pressure *P* and an axial force $F_{red}$. This configuration is referred to as the deformed configuration ℬ. The residual stress causes the cylinder wall to be under stress even if the external load is removed [25]. The stress-free reference configuration ℬ_0_ is therefore considered a rotationally symmetric cylindrical sector free of external loads, commonly referred to as the cut-open state [23] (see Figure 2). The deformation between the two configurations, ℬ_0_ and ℬ, is described by the deformation gradient **F**. Utilizing a cylindrical coordinate system with the radial, circumferential, and axial base vectors denoted, respectively, $E_{\rho }$, $E_{\phi }$, and $E_{\xi }$ in the reference configuration, and $e_{r}$, $e_{\theta }$, and $e_{z}$ in the deformed configuration, allows for the deformation gradient to be expressed as [24](1)$$F=\frac{\partial r}{\partial \rho }E_{\rho }⨂e_{r}+k\frac{r}{\rho }E_{\phi }⨂e_{\theta }+\frac{l}{\zeta }E_{\xi }⨂e_{z},$$where $k=\pi /\left(\pi -\Phi_{0}\right)$ and $\Phi_{0}\neq \pi$ is the opening angle; and ζ and *l* represent the length of the arterial segment in the referential and deformed configurations, respectively. The inequalities $\rho_{i}\leq \rho \leq \rho_{o}$ and $r_{i}\leq r\leq r_{o}$ denote the corresponding radii (see Figure 2). The indices *i* and *o* denote the inner and outer radius, respectively. For future use, we also define the wall thickness $h=r_{o}-r_{i}$ in the deformed configuration. The deformation gradient in Equation (1) is diagonal, and the principal stretches in the radial, circumferential, and axial directions are given by(2)$$\lambda_{r}=\frac{\partial r}{\partial \rho },\;\;\;\lambda_{\theta }=k\frac{r}{\rho },\;\;\;\lambda_{z}=\frac{l}{\zeta },$$
respectively. The tissue is considered incompressible, and the deformation gradient must, therefore, satisfy(3)$$\det F=\lambda_{r}\lambda_{\theta }\lambda_{z}=1.$$

Combining Equations (2) and (3) gives the following expression for the deformed radius [24]:(4)$$r\left(\rho \right)=\sqrt{r_{i}^{2}+\frac{\rho^{2}-\rho_{i}^{2}}{k\lambda_{z}}}.$$

##### Constitutive Model

The arterial wall is modeled by a Holzapfel–Gasser–Ogden (HGO) material. This model assumes an additive split of the strain–energy function into an isotropic and an anisotropic response [26]. The isotropic behavior is associated with the non-collagenous matrix and is expressed in terms of the classical neo-Hookean strain energy [27]:(5)$$\Psi^{iso}=c(I_{1}-3),$$
where c>0 is a material parameter, $I_{1}=tr\;C$ is the first invariant, and $C=F^{T}F$ is the right Cauchy–Green stretch tensor. The anisotropic response is associated with two mechanically equivalent families of collagen fibers embedded in the matrix. The two fiber families are symmetrically arranged around the circumferential direction with the pitch angle *β* in the reference configuration (see Figure 2). The collagenous material is accounted for by the strain energy.(6)$$\Psi^{aniso}=\frac{k_{1}}{2k_{2}}\left(e^{k_{2}{\left(I_{4}-1\right)}^{2}}+e^{k_{2}{\left(I_{6}-1\right)}^{2}}-2\right),$$
where $k_{1},\;k_{2}>0$ are material parameters and $I_{4}=I_{6}=\lambda_{\theta }^{2}\cos^{2}\beta +\lambda_{z}^{2}\sin^{2}\beta$ [18]. The Cauchy stress tensor σ reads [26](7)$$\sigma =-pI+\bar{\sigma },$$
where *p* is a reaction stress that arises from the incompressibility constraint in Equation (3), **I** denotes the second-order identity tensor, and $\bar{\sigma }$ is the deformation dependent part of the stress. The additive nature of the strain–energy function allows $\bar{\sigma }$ to be decomposed into two parts: an isotropic and an anisotropic stress component defined by(8)$${\bar{\sigma }}^{iso}=2F\frac{\partial \Psi_{iso}}{\partial C}F^{T},$$(9)$${\bar{\sigma }}^{aniso}=2F\frac{\partial \Psi_{aniso}}{\partial C}F^{T},$$
respectively, were $C=F^{T}F$. It is assumed that the collagen fibers only support tensile loads and buckle under compression. Therefore, the anisotropic part is omitted if $I_{4},\;I_{6}<1$. The deformation-dependent part of the stress is thus given by [26](10)$$\bar{\sigma }=\left\{\begin{array}{cc} {\bar{\sigma }}^{iso}+{\bar{\sigma }}^{aniso},\; & if\;I_{4},\;I_{6}\geq 1\\ {\bar{\sigma }}^{iso},\; & \mathrm{otherwise}. \end{array}\right.$$

##### Equilibrium and Boundary Conditions

The artery is assumed to be in a quasi-static equilibrium. The only non-trivial equilibrium equation is in the radial direction, and it reads(11)$$\frac{\partial \sigma_{rr}}{\partial r}+\frac{\sigma_{rr}-\sigma_{\theta \theta }}{r}=0.$$

Radial stress must satisfy the boundary conditions(12)$$\sigma_{rr}=\left\{\begin{array}{c} -P\;\mathrm{on}\;r=r_{i}\\ 0\;\mathrm{on}\;r=r_{o}\;, \end{array}\right.$$
which corresponds to an artery exposed to a luminal pressure *P* with zero traction on the outside. In addition, the artery is pre-stretched by a constant λ in the axial direction, resulting in a reaction force $F_{red}$, known as the reduced axial force [26].

Furthermore, the reaction stress in Equation (7) can be computed by integrating Equation (11) from $r_{i}$ to r and substituting Equations (7) and (12):(13)$$p\left(r\right)=P+{\bar{\sigma }}_{rr}\left(r\right)-\int_{r_{i}}^{r}\frac{1}{\varrho }({\bar{\sigma }}_{\theta \theta }\left(\varrho \right)-{\bar{\sigma }}_{rr}\left(\varrho \right))d\varrho .$$

#### 2.1.2. Constitutive Membrane Model

The constitutive membrane model, utilized within the in vivo parameter identification method proposed in [18], considers an artery as a homogeneous, incompressible, and thin-walled cylinder. Consequently, it is a simplified version of the arterial model described in Section 2.1.1. For this constitutive membrane model, two sets of stresses are computed: equilibrium stresses and constitutively determined stresses. Within the parameter identification method, the least-squares differences between these two sets of stresses are minimized to identify the mechanical properties of the artery in question. Here, only stresses are presented. For additional information regarding the parameter identification, the reader is referred to previous studies [13,18].

The equilibrium stresses are calculated by stating global equilibrium in the deformed configuration. Using the boundary conditions in Equation (12), the equilibrium stresses are in the circumferential direction:(14)$$\sigma_{\theta \theta }^{eq}=\left(\frac{r_{i}}{h}+\alpha \right)P$$
and in the axial direction:(15)$$\sigma_{zz}^{eq}=\frac{\pi r_{i}^{2}P+F_{red}}{\pi h\left(2r_{i}+h\right)},$$
where the superscript eq denotes quasi-static equilibrium stresses and the parameter α=0.5 is chosen to evaluate the circumferential stress for the mid-wall radius. The reduced axial force cannot be measured in vivo and is estimated by assuming that the ratio between the axial and circumferential stresses is γ=0.59 at the mean arterial pressure (MAP) $\bar{P}=13.3\;kPa$ [11]. Together with Equations (14) and (15), this yields the estimated reduced axial force(16)$${\bar{F}}_{red}=\bar{P}\pi \left[\frac{\gamma }{2}{\left(2{\bar{r}}_{i}+\bar{h}\right)}^{2}-{\bar{r}}_{i}^{2}\right],$$
where the inner radius ${\bar{r}}_{i}$ and the wall thickness $\bar{h}$ are associated with $\bar{P}$. Since the radial stress is much smaller than the membrane stresses in Equations (14) and (15), it is neglected and(17)$$\sigma_{rr}^{eq}=0.$$

The constitutively determined stresses are calculated using the constitutive model presented in Section Constitutive Model. Since the redistribution of the transmural wall stress associated with the residual stress is immaterial for membranes, the stress-free reference configuration is taken to be the deformed configuration when all external loads have been removed [18]. This unloaded configuration is denoted ℬ^∗^ (see Figure 2). In order to specify the deformation gradient $F^{mod}$ describing the deformation between ℬ^∗^ and ℬ, the circumferential and axial membrane stretches are defined in the mid-wall by(18)$$\lambda_{\theta ,m}=\frac{2r_{i}+h}{R_{i}+\sqrt{R_{i}^{2}+\lambda_{z,m}h\left(2r_{i}+h\right)}},$$(19)$$\lambda_{z,m}=\lambda ,$$
respectively, where $R_{i}$ is the inner radius in the unloaded configuration ℬ^∗^. Note that the radial stretch follows implicitly from Equation (3). The deformation gradient for the constitutive membrane model thus reads(20)$$F^{mod}=\frac{1}{\lambda_{\theta ,m}\lambda }e_{r}⨂e_{r}+\lambda_{\theta ,m}e_{\theta }⨂e_{\theta }+\lambda e_{z}⨂e_{z}.$$

By replacing the deformation gradient **F** by $F^{mod}$**,** the constitutively determined stresses are calculated using Equations (5) to (10). The reaction stress associated with the incompressibility constraint $p^{mod}$ in Equation (3) is computed using Equation (17) and reads [18](21)$$p^{mod}={\bar{\sigma }}_{rr}^{mod}=\frac{2c}{{\left(\lambda_{\theta ,m}\lambda \right)}^{2}}.$$

#### 2.1.3. Load-Bearing Fractions

The load-bearing fractions attributed to elastin and collagen are determined for both the general continuum model and the constitutive membrane model. In both cases, the fractions are calculated by integrating the respective stress component over the arterial wall and then dividing it by their sum. For instance, the load-bearing fraction attributed to collagen in the circumferential direction is defined as(22)$$\psi_{\theta \theta }^{aniso}=\frac{\int_{r_{i}}^{r_{o}}{\bar{\sigma }}_{\theta \theta }^{aniso}\left(\varrho \right)d\varrho }{\int_{r_{i}}^{r_{o}}\left({\bar{\sigma }}_{\theta \theta }^{iso}\left(\varrho \right)+{\bar{\sigma }}_{\theta \theta }^{aniso}\left(\varrho \right)\right)d\varrho }.$$

It should be noted that this calculation is straightforward for the constitutive membrane model due to the thin-walled assumption. In the following discussion, only the load-bearing fraction attributed to collagen is considered since it is assumed that the pressure load is borne solely by elastin and collagen. Consequently, the load-bearing fraction of the non-collagenous material is trivial to calculate. For example, in the circumferential direction: $\psi_{\theta \theta }^{iso}=1-\psi_{\theta \theta }^{aniso}$. The load-bearing fraction is computed for each pressure level, as defined in Section 2.2, i.e., DBP to SBP.

### 2.2. Testing Procedure and Material

Briefly, the testing procedure was as follows and is illustrated in Figure 1. A unique dataset corresponding to an aorta (Table 1) was used as an input to a thick-walled FE model (that is, an in silico model), and a pressure range (9.3–16 kPa) was applied. The inner radius $r_{i}$, the wall thickness *h*, and the wall stress states were extracted for n = 101 uniformly distributed pressure levels within the pressure range for each dataset. The pressures–radius pairs were used as input to a thin-walled model (that is, a constitutive membrane model), and a set of material parameters were identified (Table 2), along with the computation of wall stress states. The identified parameters are indicated by a superscript Id. Material parameters used for the FE model, along with wall stress states and load-bearing fraction, were compared with identified material parameters, wall stress states, and load-bearing fraction from the constitutive membrane model. Twenty one unique datasets were used for the study.

Data for this study were taken from [18] and are presented in Table 1 and Table 2, as well as from [30]. Except for the in vivo data from [30], no data were acquired in vivo for this study, including histological samples. The parameter sets in Table 1 are based on the results in [28,29] and were used to generate the in silico experiments in the form of thick-walled and residually stressed in silico abdominal aortas in Abaqus (Standard version 2017). The arterial wall is assumed to be homogeneous for the in silico arteries. This is arguably a non-physiological assumption since the aorta is a layered structure [26], but to the best of the authors’ knowledge, no similar and complete data have been published for a multi-layered wall.

The parameter sets in [18] are reorganized into the hg-, lc-, and cs-sets to study differences between groups. The hg-sets (sets 1–6) are characterized by substantial angles ($\Phi_{0}$) (see Figure 2) causing a large transmural stress gradient which increases towards the outer radius. The lc-sets (sets 7–14) are characterized by an extremely low material parameter *c* for the non-collagenous matrix and experienced large errors in the identified parameters [18]. They are kept, however, to study the effect of extreme parameter discrepancies.

The cs-sets (sets 15–21) are characterized by having a constant, or nearly constant, transmural strain distribution. The cs-sets therefore satisfy the uniform strain hypothesis at MAP proposed by [31] and are taken to represent normal healthy arteries. Datasets 15 and 16 are from [28,29]. Datasets 17–21 come from a pool of artificially created datasets and were chosen by selecting model parameters within reasonable bounds which allowed for satisfying the uniform strain hypothesis at MAP [18,30]. This expanded the parameter range investigated. It was demonstrated that the model parameters for the cs- and hg-sets are adequately identified while the parameters for the lc-sets show a much larger discrepancy [18].

In vivo human abdominal aorta data were taken from literature to form clinical datasets [30]. A subset of the cs-sets was compared to these clinical datasets for load-bearing fraction in the circumferential direction as a means of evaluation. The parameter identification method with the constitutive membrane model was used on a cohort of 30 healthy, non-smoking adult volunteers (15 males) with ages ranging from 23 of 72 years. The participants were divided into three age-groups, with five males and five females in each group [30].

### 2.3. Statistics

The stress state in the in silico aorta varies through the thickness. To enable a comparison with the membrane model, the wall stresses are computed at mid-wall. To assess the agreement between the predicted stress state in the constitutive membrane model and the stress state in the in silico model, we use two different measures: the maximum difference $\Delta_{max}$ and the coefficient of determination, ${\hat{R}}^{2}$. Let $y_{k}^{Pre}$ and $y_{k}$ denote components of the stress state vector in the constitutive membrane model and in the in silico model, respectively. The coefficient of determination is defined by [32](23)$${\hat{R}}^{2}=1-\frac{\sum_{k=1}^{n}{\left(-\Delta_{k}+\hat{\Delta }\right)}^{2}}{\sum_{k=1}^{n}{\left(y_{k}-\hat{y}\right)}^{2}},$$
where $\Delta_{k}=y_{k}^{Pre}-y_{k}$, $\hat{\Delta }$, and $\hat{y}$ denote the mean of Δ and *y*, respectively, and the index k=1,…, n represents a pressure level. The distinction between ${\hat{R}}^{2}$ and its standard definition $R^{2}$ is that the former is not affected by an offset between two curves. This offset is instead accounted for in the maximum difference (see Discussion). In some cases, ${\hat{R}}^{2}$ is negative, indicating a complete lack of agreement [32], and is then set to zero to facilitate the interpretation. The same two measures are used to compare the load-bearing fraction attributed to collagen between the constitutive membrane model and the in silico model.

The transmural stress gradient (the stress variation through the wall thickness) for the in silico aortas was computed by subtracting the stress at the inner wall (minimum) from the stress at the outer wall (maximum). Since the transmural gradient increases with blood pressure, the SBP of 16 kPa was used for the transmural gradient computation.

Differences between the three sets of arteries are tested using the non-parametric Kruskal–Wallis test [33], with the significance level for the test chosen to *p =* 0.05. In cases of significant differences, two-sample Mann–Whitney–Wilcoxon tests [34] with corrected significance level according to Bonferroni [35] are performed. Results are only provided for the Mann–Whitney–Wilcoxon test in cases where one set is significantly different compared with both other sets. The highest *p*-value is stated in all cases.

Data for males and females from [30] were pooled into one group, which included all study participants [36]. A comparison of the load-bearing fraction attributed to collagen in the circumferential direction among the clinical datasets, cs-sets, hg-sets, and lc-sets was performed using a one-way ANOVA test. A Bonferroni correction was employed for a pairwise comparison of the groups [35]. The fractions were compared at both SBP and DBP. The ANOVA test was used since the clinical datasets were reported as mean and standard error of the mean [30].

### 2.4. Software

Abaqus (Standard version 6.12–3) was utilized for the simulations of the FE model. MATLAB (The Mathwork, Natick, MA, USA) version 8.4 (R2014b) was employed for computation. IBM SPSS Statistics Version 27 (IBM Corporation, Somers, NY, USA) was used for statistical analysis.

## 3. Results

The isotropic and anisotropic stress components, the total stress, the reaction stress, and the load-bearing fraction attributed to collagen were compared between the constitutive membrane model and the in silico artery. The calculated metrics are summarized separately in Table 3, Table 4 and Table 5.

In the circumferential direction, the cs-sets exhibited significantly lower values than the hg- and lc-sets in terms of both the total stress (*p =* 0.013) and the anisotropic stress component (*p =* 0.013) (see Table 3). The shape agreement, as indicated by ${\hat{R}}^{2}$, was also significantly better for the total stress and the anisotropic stress component (*p =* 0.028 and *p =* 0.010, respectively). For the isotropic stress component, there was no significant difference between the cs-, hg-, and lc-sets. The shape agreement for the lc-sets, however, was significantly lower (*p =* 0.011) compared with the cs- and hg-sets.

The transmural stress gradient (median, interquartile range) exhibited a significantly lower median-value (*p =* 0.008) for the cs-sets (19.10, 16.86–31.03 kPa) compared to the hg- and lc-sets (429.00, 341.81–529.07 kPa and 520.43, 248.43–758.57 kPa, respectively). This is illustrated in Figure 3. The cs-set demonstrated an almost linear transmural stress gradient (panel B, black dashed line), while the hg-set displayed a non-linear transmural gradient (panel A, black dashed line). The Pearson correlation coefficient indicated an increasing maximum difference ($\Delta_{max}$) with increasing magnitude of the transmural stress gradient (*r* > 0.99, *p <* 1 × 10^−17^) for the cs-, hg-, and lc-sets.

In the axial direction, the difference in the isotropic stress component was significantly (*p =* 0.036) larger for the lc-sets compared with the cs- and hg-sets. For the total stress, the maximum difference for the cs-sets was significantly lower (*p =* 0.043) and the shape agreement was significantly higher (*p =* 0.022) when compared with the lc- and hg-sets. For the anisotropic stress component, there was no significant difference between the cs-, hg- and lc-sets (see Table 4).

The transmural stress gradient in the axial direction behaves similarly to that in the circumferential direction. This is illustrated in Figure 4. The stress gradient for the cs-sets (14.09, 13.29–21.62 kPa) exhibited a significantly (*p =* 0.01) lower median value compared with the hg-sets and lc-sets (216.24, 173.94–267.87 kPa and 263.48, 129.17–361.95 kPa, respectively). The cs-set was almost linear throughout the arterial wall (panel B, black dashed line), while the hg-set showed a non-linear transmural gradient (panel A, black dashed line). This was also true for the lc-set.

The load-bearing fraction attributed to collagen in the circumferential direction seemed to be unaffected by the transmural stress gradient and was well predicted in all cases. This is evident as the cs-set (low transmural stress gradient) and hg-set (high transmural stress gradient) exhibited similar values, while demonstrating a significant difference compared to the lc-set (Table 3).

In terms of the load-bearing fraction attributed to collagen, there was almost no difference in the circumferential direction between the predicted value in the constitutive membrane model and the corresponding in silico model for the three sets (see Table 3). In the axial direction, larger differences were observed, particularly in the lc-sets (see Table 4). The shape agreement, i.e., how the load-bearing fraction attributed to collagen changes with increasing blood pressure, was significantly worse for the lc-sets in both the circumferential (*p =* 0.006) and axial direction (*p =* 0.019) when compared with the cs- and hg-sets.

The cs-sets predicted a load-bearing fraction attributed to collagen in the circumferential direction for both the constitutive membrane model and the in silico artery (Table 3 and Table 4) of approximately 22% at systolic blood pressure (SBP) and 10% at diastolic blood pressure (DBP) (Figure 5); in the axial direction, it was approximately 18% at SBP and 8% at DBP.

There was no significant difference in the reaction stress between the three sets (see Table 5).

The load-bearing fraction attributed to collagen in circumferential direction is displayed in Figure 5. When comparing the cs-sets and the clinical datasets, the load-bearing fraction attributed to collagen in the circumferential direction exhibited good agreement, as well as a similar behavior with respect to pressure. The load-bearing fraction in the clinical datasets was 1.4-fold higher compared to the cs-sets for SBP (dark grey bars), as well as DBP (light grey bars). However, the load-bearing fraction attributed to collagen of the cs-sets displayed no significant difference compared to the clinical datasets at SBP (*p =* 0.79) and DBP (*p =* 1.00). At SBP, both the hg-sets and the lc-sets exhibited significant higher values for the load-bearing fraction compared to the clinical datasets (155% *p* < 0.05, 210% *p* < 0.05) and cs-sets (260% *p* < 0.05, 350% *p* < 0.05). At DBP, the differences were even more pronounced when comparing the hg-sets and the lc-sets to clinical datasets (410% *p* < 0.05, 620% *p* < 0.05) and cs-sets (620% *p* < 0.05, 900% *p* < 0.05). In addition, the lc-sets demonstrated a higher value compared to the hg-sets at both SBP (20%) and DBP (40%) (*p* < 0.05 for both). The load-bearing fraction at SBP was 2.3 times higher than at DBP for both the cs-sets and the clinical datasets. In contrast, it was only 1.2-fold higher for the hg-sats. For the lc-sets, there was no difference between SBP and DBP.

## 4. Discussion

The main findings of this study are as follows:(1)For an in silico aorta exhibiting a physiological transmural stress gradient, the constitutive membrane model’s total stress, as well as its isotropic and anisotropic components in the circumferential direction, are well predicted. In the axial direction, the prediction capability is good, although the lack of information regarding the axial loading condition in the in vivo measurements sets a limit for the accuracy.(2)The accuracy of the predicted stress state in both the circumferential and axial directions deteriorate with increasing transmural stress gradient. This is explained by the assumption of a thin vascular wall in the in vivo parameter identification method.(3)The load-bearing fraction attributed to collagen is well predicted for all blood pressure levels, particularly in the circumferential direction, and is independent of the transmural stress gradient.(4)The results for the load-bearing fraction attributed to collagen in the circumferential direction, compared with previously published data from in vivo human abdominal aorta, demonstrate similar behavior with respect to pressure and are of equal magnitude. However, in vivo measurements are somewhat higher. This discrepancy may, in part, be explained by the variation in age of the reference (in vivo) population, which, ranges from young to elderly males as well as females.

Table 3 and Table 4 demonstrate that the stress state is significantly better reproduced for the cs-sets with low values for the maximal difference ($\Delta_{max}$), while the coefficient of determination (${\hat{R}}^{2}$) has a value of 1 or very close to 1, compared with the hg- and lc-sets. The results reveal no significant difference between the hg- and lc-sets. Since the parameters for the hg-sets are adequately identified [18], this suggests that the parameter discrepancy does not introduce a large error in the stress state. The superior results for the cs-sets must, therefore, be attributed to another factor.

For the hg- and lc-sets, the circumferential total stress is significantly overpredicted in the mid- wall (see Table 3). The discrepancy is primarily due to the transmural stress gradient of the in silico abdominal aortas. The transmural stress gradient is considerably larger for the hg- and lc-sets compared with the cs-sets. Additionally, the maximum overprediction for the hg- and lc-sets increases with an increasing transmural stress gradient, as indicated by the Pearson correlation coefficient. This observation is reasonable, as the membrane approximation becomes progressively less correct as transmural gradients increase. In the case of the cs-sets, which exhibit a low transmural gradient, the membrane approximation is appropriate, and the maximum difference is notably close to 8 kPa. Consequently, the difference, as it manifests at the systolic pressure of 16 kPa (SBP), can be explained by the choice of α=0.5 in Equation (14) for the evaluation of stress at the mid-wall radius. This results in an overprediction of approximately 8 kPa. Owing to the transmural stress gradient, this overprediction is correlated with the chosen location of the radius in the thick-walled constitutive model. The selected location will determine the value of the transmural stress from the thick-walled model, which is then compared to the constitutive membrane model. Choosing a position close to the outer radius of the vessel results in a lower value for the overprediction. Selecting a radius closer to the luminal side (inner radius) leads to a better agreement of the identified parameters when comparing the thick-walled constitutive model to the constitutive membrane model. Utilizing a mid-wall radius (α=0.5) is a compromise intended to achieve an acceptable agreement of model parameters and to limit the overprediction. While the constitutive membrane model produces adequate results for the cs-sets due to a low transmural stress gradient throughout the cardiac cycle, it does not yield good results for the hg- and lc-sets, which exhibit a large transmural stress gradient (see Figure 3).

In contrast, the load-bearing fraction attributed to collagen in the circumferential direction appears to be unaffected by the transmural stress gradient and is well predicted in all cases (see the cs-set and hg-set in Table 3). This can be explained by the definition of the load-bearing fraction in Equation (22), which effectively averages the varying stress components over the arterial wall. Consequently, the transmural gradient does not negatively impact the predicted fraction.

During the study, it was observed that evaluation of the membrane stresses in Equation (14), using the mid-wall radius, positively influenced parameter identification. This resulted in an overestimation of the total circumferential stress by approximately 8 kPa, which is related to the reaction stress. The difference between the reaction stresses in the constitutive membrane model (thin-walled model) and in the general continuum model (thick-walled model) can be computed using Equations (13) and (21), yielding the following:(24)$$p^{mod}-p\left(r\right)={\bar{\sigma }}_{rr}^{mod}-{\bar{\sigma }}_{rr}\left(r\right)-P+\int_{r_{i}}^{r}\frac{1}{\varrho }(\sigma_{\theta \theta }\left(\varrho \right)-\sigma_{rr}\left(\varrho \right))d\varrho .$$

If the true model parameters in Table 1 are used, the isochoric radial stress coincides in both models at the mid-wall, that is ${\bar{\sigma }}_{rr}^{mod}={\bar{\sigma }}_{rr}\left(r_{m}\right)$. It is important to note that the identified parameters in Table 2 differ from those defined for the in silico arteries, and the predicted isochoric radial stress ${\bar{\sigma }}_{rr}^{Pre}$ is only approximately equal to ${\bar{\sigma }}_{rr}$ at the mid-wall.

Since the integral in Equation (24) varies from zero to *P*, the right-hand side will always be negative, and it can be concluded that the reaction stress is systematically underpredicted in the constitutive membrane model due to the thin-walled assumption. By overestimating the total circumferential stress by approximately 8 kPa, this systematic underprediction in the reaction stress is compensated for. As a consequence, the isotropic and anisotropic stress components are well predicted in the circumferential direction, particularly for the cs-sets, and the load-bearing fraction attributed to collagen is also well predicted for all sets.

In the circumferential direction, the axial stress used in the parameter identification is calculated from the equilibrium equation (Equation (15)). This axial equilibrium stress is inaccurate for two reasons: first, an incorrect reduced axial force ${\bar{F}}_{red}$ is employed since the correct force cannot be measured in vivo; second, a stress gradient is present within the arterial wall. The correct reduced axial force and the estimated reduced axial force are included in Table 1 and Table 2, respectively. It is apparent that the transmural stress gradient in the axial direction exhibits similar behavior to that in the circumferential direction (see Figure 4). The stress gradient for the cs-sets is significantly smaller compared with the hg- and lc-sets and is nearly linear throughout the arterial wall. In contrast, both the hg- and lc-sets display a markedly non-linear transmural gradient of high magnitude.

With respect to the load-bearing fraction attributed to collagen in the axial direction, larger differences are observed compared to the circumferential direction. The maximum difference, $\Delta_{max}$, for the cs-sets is 0.027 in the axial direction and 0.01 in the circumferential direction (Table 3 and Table 4). Although the load-bearing fraction is not significantly affected by large transmural gradients, the incorrectly reduced axial force and the resulting erroneously identified axial pre-stretch limit a more precise agreement.

Both the hg- and lc-sets exhibit high transmural stress gradients in the circumferential and axial directions (see Figure 3 and Figure 4). This observation prompts the question of the physiological accuracy of these in silico abdominal aortas, particularly in the context of homeostasis-driven growth and remodeling [6,37,38], where sustained gradients of several 100 kPa appear highly unphysiological. This issue is also highlighted in [29], where the authors propose that the gradient results from a stress redistribution towards the adventitia to shield the intima from high stresses. This explanation seems unlikely since the adventitia is believed to be a protective sheath, shielding the artery from acute overextension, rather than being the primary load-carrying layer [24,26,39]. We hypothesize that the cause of the high transmural stress gradients lies in the representation of residual stress, specifically, the opening angle method ($\Phi_{0}$) (see Figure 2 and Equation (1)). It is straightforward to show that a necessary condition for a uniform strain distribution in a homogeneous cylinder is that the opening angle $\Phi_{0}<\pi$ (see Appendix A). As evident from Table 1, more than half of the arteries in [29] do not meet this requirement, leading to a radially growing transmural stress gradient that increases with pressure. The complexity of the residual stress fields in these arteries exceeds the representational capacity of a single opening angle [40,41].

The parameter γ was investigated with respect to the influence on the identification of parameters [18]. Their findings indicated that using an incorrect value may lead to suboptimal parameter estimation. The proposed value of γ=0.59 at MAP fell within the reported range across the three different datasets, approximately 0.47–0.71 [18].

The agreement between the predicted and in silico stress components is analyzed using the maximum difference, $\Delta_{max}$, and the coefficient of determination, ${\hat{R}}^{2}$. While the interpretation of the maximum difference is straightforward, the coefficient of determination may require some explanation. When comparing ${\hat{R}}^{2}$ with standard $R^{2}$, the former is corrected by the mean difference of the offset of the two curves [32]. This is illustrated for the circumferential total stress for sets 4 and 5 in Figure 6. Comparing the solid and dashed lines will produce standard $R^{2}$, which will be affected by both the curve shape (here represented by the slopes of the lines) and the offset, which is clearly visible in Figure 6. Modified ${\hat{R}}^{2}$ comes from comparing the dotted and the dashed lines. Visually, there is no obvious offset, although curve shape still differs. Instead, the offset is represented by the maximum difference, $\Delta_{max}$. Since $R^{2}$ relates to goodness of fit in regression analysis, ${\hat{R}}^{2}$ is used in the same way, but with the limitation of only quantifying the goodness of the shape agreement. Other measures, such as $R^{2}$ and the modal assurance criterion [42], have also been considered, but they were not sensitive enough.

Separating the stress state into an isotropic and an anisotropic component may have physiological implications. There is compelling evidence in the literature that arteries constantly adapt to their mechanical environment in order to maintain homeostasis, e.g., wall thickening and an increased collagen synthesis associated with hypertension [6,19,21,22,43] or the increased deposition of collagen associated with aneurysm growth [44]. Observations like these have led to the hypothesis that growth and remodeling of the vascular wall is driven by mechanical factors such as the stress state [42,45,46,47,48]. The arterial low-pressure response is essentially controlled by elastic lamellae primarily built from isotropic elastin sheets [49,50,51]. The high-pressure response on the other hand is linked to the collagen networks [51,52,53]. Based on histology, elastin and collagen appear to be the major wall constituents for the isotropic and anisotropic stress components, respectively. It may therefore be possible to relate the analysis of stress components to the presence of wall constituents and their load-bearing fraction to growth and remodeling.

This is further illustrated by the analysis of the load-bearing fractions attributed to collagen in the circumferential direction, as predicted from the constitutive membrane model using the cs-, hg-, and lc sets, as well as clinical data from the literature [30]. The constitutive membrane model, when applied to the cs-sets and hg-sets, displays a higher load-bearing fraction at SBP compared to DBP. This is consistent with the known behavior of collagen, which is gradually recruited as a load-bearer with increasing blood pressure [53]. The lc-sets show no difference at SBP compared with DBP, which deviates from the expected physiological behavior. At pressures below physiological blood pressure, collagen is wavy and unstretched. As the load increases, collagen stretches until the fibers bear the load and are fully engaged at high physiological blood pressure [53]. Furthermore, the helical nature of collagen distribution in the arterial wall results in an increased circular and decreased longitudinal orientation of the collagen fibers with increasing distention of the vessel (e.g., due to increased blood pressure). In the context of a pressure-related increase in the load-bearing fraction, this may indicate increased stretch of circumferentially oriented collagen bundles.

The magnitude of the load-bearing fraction attributed to collagen in the circumferential direction exhibits a lower value for both the cs- and clinical datasets compared to the hg- and lc-sets. There is no significant difference observed between the cs- and clinical datasets. However, the values for the lc-sets are significantly higher compared to the hg-sets. These elevated values may not represent physiologically relevant states. Both the lc- and the hg-sets appear to possess properties the fall outside the expected physiological range.

The cs-sets, specifically numbers 17, 18, 19, 20, and 21, exhibited curve shapes for the stress state and its components that were analogous to those observed in the clinical datasets. The remaining sets (numbers 1–16) demonstrated significant deviations from the clinical datasets in terms of the curve shapes of stress and its components. This atypical behavior of the in silico aortas, derived from the hg- and lc-sets, may account, at least partially, for the differences reported in the comparative analysis of the four datasets.

The parameter identification method was employed on a small cohort to investigate the human abdominal aorta [30]. The same behavior was demonstrated when comparing the small cohort to the cs-sets, with the load-bearing fraction attributed to collagen in the circumferential direction increasing with increasing pressure. There was no observed difference between males and females at DBP and SBP. Furthermore, in males, they demonstrated a decrease in the load-bearing fraction attributed to collagen in the circumferential direction with increasing age at DBP and what appeared to be a decrease with age at SBP (*p =* 0.06). In females, the load-bearing fraction attributed to collagen did not change with age, either at SBP or DBP [30]. In the context of an age-related decrease in the load-bearing fraction attributable to collagen, this may indicate an increased structural and chemical alteration of collagen [54,55].

Arterial wall constituents may undergo temporal changes, exhibiting a decrease in concentration but preserved elastin content, and an increase in both content and concentration of collagen [55]. Structural damage to lamellar units, characterized by thinning, splitting, and fraying, may impact elastin. Collagen shows structural changes, with an increase in the random distribution of layers within the arterial wall and specific layers. Additionally, the subendothelial layer of the intima appears to thicken due to dispersed collagen, resulting from age-related increases in vessel radius. Furthermore, collagen fibers seem to adopt a more circumferential orientation within the media, attributed to the stretching of the network of helices. Chemical alterations, including elastin glycation and collagen fiber cross-linking, also occur with advancing age [54,55]. These combined structural and chemical changes may suggest that age-related modifications in collagen could adversely affect its stiffness and ability to withstand high loads.

The cs-sets and the clinical datasets exhibit an approximately equal increase in load-bearing fraction in the circumferential direction from DBP to SBP. However, the clinical datasets display marginally higher values compared to the cs-sets. The representation of sex and age in the clinical datasets may offer some explanation to this observation. The clinical datasets consist of males and females as well as various age groups, while the cs-sets have been designed without the consideration of sex and age. Consequently, the cs-sets cannot reveal a variation in load-bearing fraction with sex and age. In addition, the number of observations differs by a factor of six between cs-sets (n = 5) and clinical datasets (n = 30). Nevertheless, the resulting load-bearing fraction attributed to collagen in the circumferential direction demonstrated no significant difference when comparing the cs-sets with the clinical datasets for SBP (10 percentage points) as well as DBP (5 percentage points) (Figure 5).

Since the load-bearing fraction attributed to collagen in the circumferential direction is well predicted when the constitutive membrane model is compared to the FE model, and the cs-sets display good agreement with the clinical datasets, the predicted capability is believed to be at least acceptable. However, this needs to be corroborated by further research.

The load-bearing fraction, which is of similar magnitude in the cs-sets and in the clinical datasets, further supports the potential use of the parameter identification method for studying both isotropic and anisotropic vessel wall properties, and for possible patient care. This analysis provides knowledge of the mechanical properties of the arterial wall that can be added to clinical applications such as the pressure–strain elastic modulus (Ep) [7], the stiffness beta-index [8], and the pulse wave velocity [9], all of which reflect overall vessel wall stiffness. Pathologies such as aortic aneurysm, hypertension, Ehlers–Danlos syndrome, and Marfan syndrome all exhibit potential changes to the constituents of the vascular wall. Age-related alterations, as well as structural and chemical modifications of the vascular wall, may be detectable through changes in load-bearing fraction attributed to collagen.

A comprehensive model of the vasculature should consider both passive and active vessel wall behavior in response to forces initiated by blood pressure [56]. Different regions of the vasculature have varying constituents that dictate the regional response to external forces. The aorta is primarily elastic and displays mainly passive behavior. In contrast, the femoral artery is a muscular vessel with a wall rich in contracting smooth muscle cells, necessitating the consideration of active behavior. Further proximal in the vascular tree, smaller vessels interact strongly with the vascular endothelium, and depending on the activation mechanism of contraction initiation, the mechanical response can vary [53]. For these regions, a passive model may not suffice, and a model including active behavior may be required. However, the abdominal aorta has a scarcity of contracting smooth muscle cells [53]. Therefore, a passive model may be adequate to study the mechanical properties of the abdominal aortic vessel wall [56].

This study focuses on the evaluation of the suggested parameter identification method rather than assessing the predictive capability of the constitutive model. While the model’s predictive capability is important, the efficacy of the parameter identification process is equally significant. Identification of the global minimum in nonconvex problems and ill-conditioned error-function Hessians is crucial for accurate parameter identification [57].

Ambiguity in the identified parameters associated with the HGO-model has raised concern [58]. The performance of the constitutive model in terms of precision and accuracy has been studied, suggesting the model has both strengths and weaknesses, although specific questions regarding parameter identification were not considered [59,60]. Therefore, alternative approaches should be explored to ensure correct parameter identification [14,15,17,57].

The challenges of multiple minimums or finding a global minimum for a fitting solution, as well as overparameterization, are two of the most common difficulties associated with parameter identification. The task of finding the minimum can be addressed by initiating the fitting process from multiple starting points and selecting the solution with the smallest minimum. Overparameterization means that more than one parameter combination can solve the minimization problem, making it difficult to determine which parameter combination best represents the mechanical properties of the artery. This issue may be addressed by simplifying the constitutive model, i.e., using fewer parameters or using fixed model parameters, or increasing the amount of data collected [14,15,17,57].

## 5. Conclusions

This study assesses the accuracy of the stress state and of the load-bearing fraction attributed to collagen, as predicted by an in vivo parameter identification method using in silico finite element models. We have demonstrated that at a physiological transmural stress gradient, total stress, as well as its isotropic and anisotropic components, are accurately predicted in the circumferential direction, as indicated by the value for $\Delta_{max}$ and ${\hat{R}}^{2}$. In the axial direction, somewhat larger differences occur, which are related to the lack of information about the axial loading conditions in the in vivo measurement. The accuracy of the predicted stress state in both the circumferential and the axial direction deteriorates with increasing transmural stress gradient, which is explained by the assumption of a thin wall in the in vivo parameter identification method. Regardless of the transmural stress gradient, the load-bearing fraction attributed to collagen is well predicted for the blood pressure range 9.3–16.0 kPa (approximately 70–120 mmHg), particularly in the circumferential direction.

## Figures and Tables

**Figure 1 medsci-13-00009-f001:**
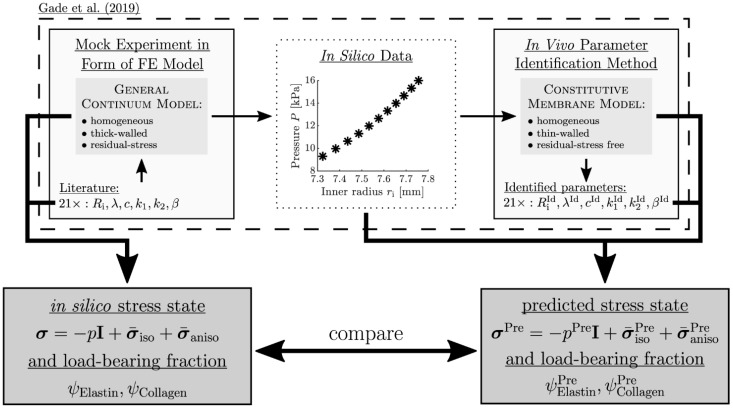
Methodology for obtaining the stress state and load-bearing fractions of the in silico experiments and their prediction through the in vivo parameter identification method. The contents of the dashed box are part of [18] and the thick black arrows and the dark grey boxes are part of this paper. For an explanation of the variables, see Section 2.1.

**Figure 2 medsci-13-00009-f002:**
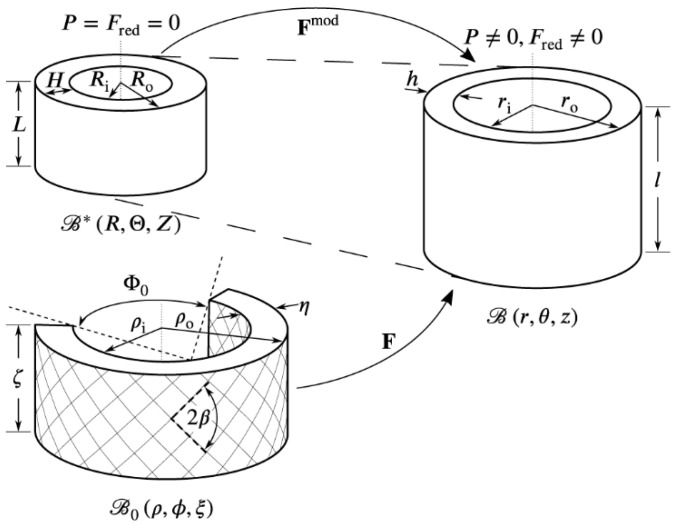
Stress-free (ℬ_0_), unloaded (ℬ^∗^), and deformed (ℬ) configuration of an arterial segment. A description of the parameters can be found in Section 2.

**Figure 3 medsci-13-00009-f003:**
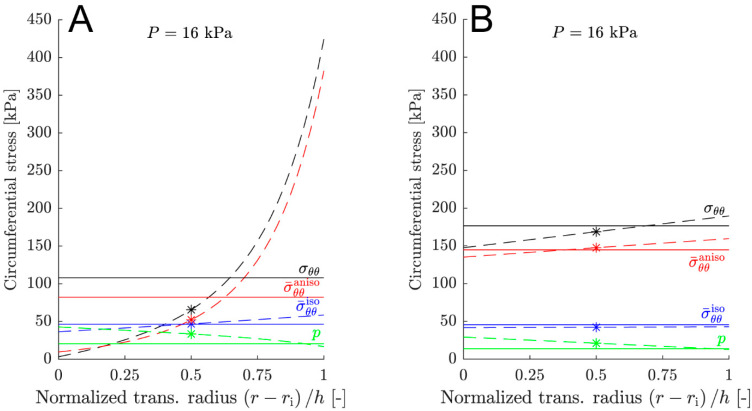
Circumferential transmural stress state at the systolic pressure for parameter sets 4 (**A**) and 15 (**B**). Set 4 represents a high transmural stress gradient, while set 15 represents a low transmural stress gradient. The solid lines depict the predicted values by the constitutive membrane model (see Table 2). The dashed lines correspond to the stress state of the in silico abdominal aorta and the asterisks denote the stress state in the mid-wall of the in silico abdominal aorta (see Table 1). The color black represents the total stress $\sigma_{\theta \theta }$, red signifies the anisotropic stress ${\bar{\sigma }}_{\theta \theta }^{aniso}$, blue denotes the isotropic stress ${\bar{\sigma }}_{\theta \theta }^{iso}$, and green indicates the reaction stress *p*. The subscript *θθ* specifies circumferential direction.

**Figure 4 medsci-13-00009-f004:**
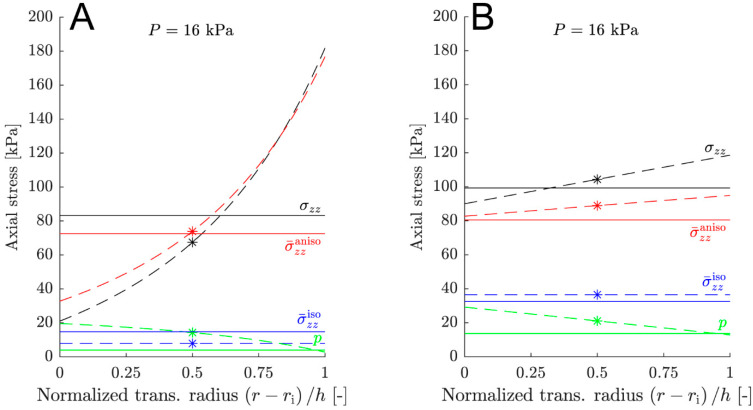
Axial transmural stress state at systole for parameter sets 5 (**A**) and 15 (**B**). Set 5 represents a high transmural stress gradient, while set 15 represents a low transmural stress gradient. The solid lines depict the predicted values by the constitutive membrane model (see Table 2). The dashed lines correspond to the stress state of the in silico abdominal aorta, and the asterisks denote the stress state in the mid-wall of the in silico abdominal aorta (see Table 1). The colors represent different types of stress: black for the total stress $\sigma_{zz}$, red for the anisotropic stress ${\bar{\sigma }}_{zz}^{aniso}$, blue for the isotropic stress ${\bar{\sigma }}_{zz}^{iso}$, and green for the reaction stress *p*. The subscript *zz* specifies axial direction.

**Figure 5 medsci-13-00009-f005:**
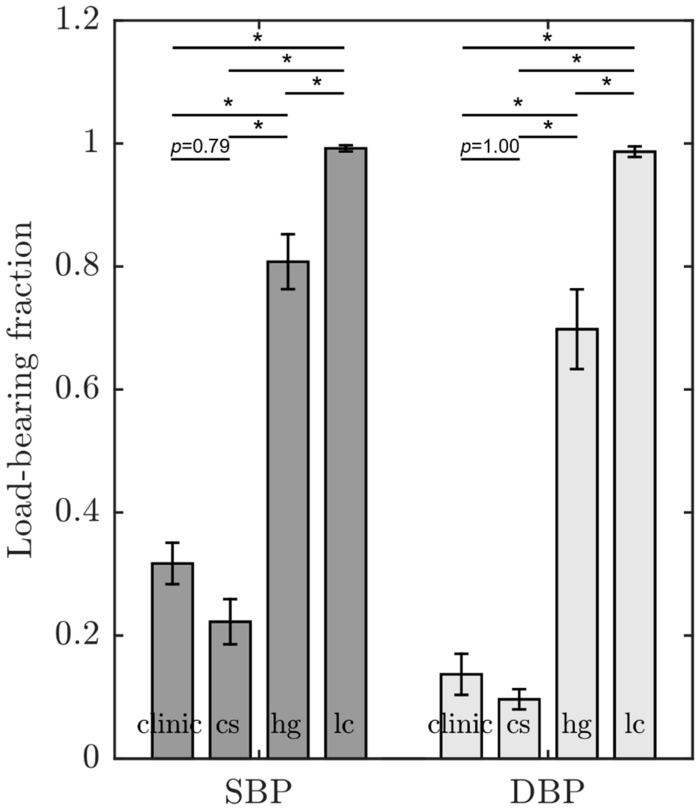
Load-bearing fraction attributed to collagen in circumferential direction, as derived from clinical datasets and cs-, hg-, and lc-sets at SBP (dark grey) and DBP (light grey). The clinical datasets were taken from the literature [30]. The load-bearing fraction was computed using the parameter identification method with the constitutive membrane model. Results are presented as mean ± SEM. clinic: clinical datasets; cs: cs-sets (numbers 17–21); hg: hg-sets; lc: lc-sets. *: *p* < 0.05.

**Figure 6 medsci-13-00009-f006:**
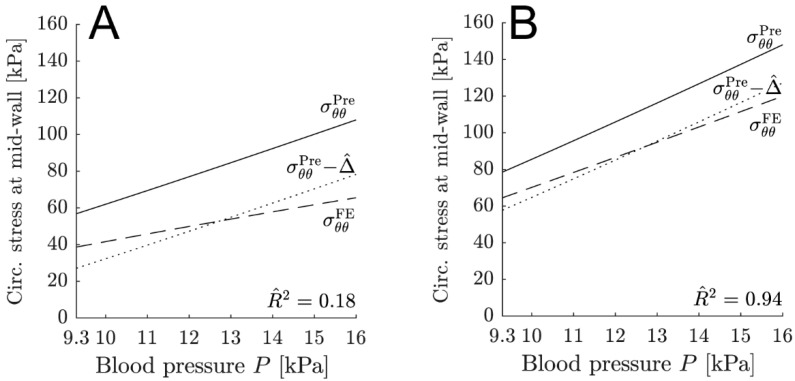
Circumferential total stress state during a cardiac cycle for parameter sets 4 (**A**) and 5 (**B**). For illustrative purposes, sets 4 and 5 are shown. The sets represent a high transmural stress gradient, although with different ${\hat{R}}^{2}$. The dashed line represents the stress state $\sigma_{\theta \theta }^{FE}$ in the mid-wall of the in silico abdominal aorta (see Table 1). The solid line is associated with the predicted value $\sigma_{\theta \theta }^{Pre}$ (see Table 2), and the dotted line depicts the predicted stress state corrected by the mean difference $\hat{\Delta }$ between the predicted and in silico stress state. Note that the values for $\sigma_{\theta \theta }^{FE}$ and $\sigma_{\theta \theta }^{Pre}$ at 16 kPa correspond to the black asterisk and the black solid line in Figure 3, respectively. The subscript *θθ* denotes the circumferential direction.

**Table 1 medsci-13-00009-t001:** Parameter sets of the in silico abdominal aortas. The set numbers in parenthesis correspond to the naming convention in [28]. The sets numbered 1–16 are derived from [29] and supplemented with data from [18,28]. The sets numbered 17–21 are sourced from [18]. Note that the reduced axial force is stated as its mean value ${\hat{F}}_{red}$ of the corresponding FE model.

		Labrosse			Horný				Gade	
		R	H	$$\Phi_{0}$$	c	$$k_{1}$$	$$k_{2}$$	β	λ	$${\hat{F}}_{red}$$
	Set	[mm]	[mm]	[deg]	[kPa]	[kPa]	[-]	[deg]	[-]	[N]
hg-sets	1 (F49)	5.9	1.51	252.00	2.31	20.06	4.11	39.95	1.1699	0.57
	2 (M60)	6.3	1.69	156.00	5.91	17.57	3.18	43.16	1.1217	0.83
	3 (M61b)	7.3	1.62	335.00	6.98	11.37	8.05	41.95	1.1039	0.67
	4 (M66)	7.2	1.78	253.00	17.78	25.72	7.65	38.85	1.1385	1.49
	5 (M70a)	7.1	1.23	208.00	2.78	67.45	7.02	35.49	1.1908	0.60
	6 (M70b)	7.4	1.64	201.00	12.18	17.31	22.86	39.69	1.0582	0.52
lc-sets	7 (F50)	6.7	1.14	323.00	0.68	49.54	7.44	36.67	1.1791	0.41
	8 (F65)	6.2	1.21	248.00	0.05	9.06	5.87	39.36	1.1932	0.36
	9 (M57)	7.5	1.28	322.00	0.05	4.37	4.87	41.11	1.2080	0.55
	10 (M61a)	7.7	1.22	270.00	0.05	49.45	26.74	37.55	1.0836	0.25
	11 (M67a)	8.0	1.58	118.00	0.05	41.95	22.00	38.18	1.0666	0.34
	12 (M67b)	7.9	1.26	174.00	0.05	61.02	7.02	35.49	1.1984	0.56
	13 (M71)	10.0	1.72	118.00	0.05	94.49	9.26	37.65	1.0964	0.84
	14 (M77)	7.0	1.50	135.00	0.05	26.65	39.30	38.23	1.0500	0.19
cs-sets	15 (F63)	5.4	0.96	96.00	13.59	41.78	3.29	39.86	1.1594	0.74
	16 (M38)	5.3	1.22	117.00	12.20	19.28	3.22	41.60	1.1576	0.74
	17	10.1	2.81	46.19	32.68	7.44	47.90	40.02	1.0210	0.65
	18	14.9	2.81	52.23	67.39	22.53	11.79	47.47	1.0489	5.95
	19	5.3	1.41	20.98	75.91	71.16	67.08	37.06	1.0422	1.03
	20	3.9	0.68	28.81	90.87	64.35	38.12	39.33	1.0281	0.29
	21	12.3	3.54	81.75	28.89	21.33	3.29	53.24	1.0372	3.18

**Table 2 medsci-13-00009-t002:** Parameter sets for the constitutive membrane model are derived from [18]. The superscript Id denotes identified and ${\hat{F}}_{red}$ denotes the estimated constant reduced axial force used in the parameter identification.

		$$R_{i}^{Id}$$	$$c^{Id}$$	$$k_{1}^{Id}$$	$$k_{2}^{Id}$$	$$\beta^{Id}$$	$$\lambda^{Id}$$	$${\bar{F}}_{red}$$
	Set	[mm]	[kPa]	[kPa]	[-]	[deg]	[-]	[N]
hg-sets	1 (F49)	6.3	3.53	21.38	4.24	31.25	1.3675	0.85
	2 (M60)	6.6	6.84	16.70	2.92	38.63	1.1948	1.13
	3 (M61b)	7.7	7.88	16.69	10.23	32.94	1.2400	1.20
	4 (M66)	7.0	16.85	33.98	8.47	37.11	1.0918	1.13
	5 (M70a)	7.3	4.04	67.00	6.82	29.74	1.3412	1.86
	6 (M70b)	7.7	11.69	25.54	24.08	32.67	1.1644	1.17
lc-sets	7 (F50)	7.2	2.15	53.94	8.27	23.21	1.6869	0.77
	8 (F65)	6.7	0.32	10.84	6.25	9.56	4.3363	0.84
	9 (M57)	8.8	0.39	7.93	6.59	9.67	4.3654	1.26
	10 (M61a)	7.2	0.19	75.80	32.36	6.01	5.8890	0.98
	11 (M67a)	7.5	0.18	46.77	20.74	6.46	5.5922	1.27
	12 (M67b)	7.9	0.18	57.97	6.51	7.85	4.9535	1.05
	13 (M71)	9.9	0.16	87.93	8.39	7.08	5.2656	1.80
	14 (M77)	6.0	0.29	5.93	22.21	8.08	4.6043	1.00
cs-sets	15 (F63)	5.2	13.64	41.97	3.25	41.43	1.0918	0.58
	16 (M38)	5.2	12.26	19.55	3.09	41.44	1.1272	0.69
	17	10.3	35.33	6.42	43.22	35.87	1.0519	2.58
	18	14.8	66.37	21.16	10.46	45.55	1.0290	4.46
	19	5.3	74.78	61.92	57.51	33.51	1.0211	0.62
	20	3.9	91.05	57.02	33.92	36.71	1.0215	0.27
	21	12.4	29.58	17.53	2.90	51.18	1.0457	4.57
Clinic *		7.12	102.70	8.35	150.65	42.35	1.042	-

*: Clinic: the clinical datasets comprise 30 subjects, both females and males, age 23–72 years, taken from [30].

**Table 3 medsci-13-00009-t003:** Maximum difference ($\Delta_{max}$) and coefficient of determination (${\hat{R}}^{2}$) of predicted and in silico circumferential measures for each parameter set.

	CIRCUMFERENTIAL DIRECTION								
		$$\mathrm{Isotropic}\;\mathrm{Stress}\;{\bar{\sigma }}_{\theta \theta }^{iso}$$		$$\mathrm{Anisotropic}\;\mathrm{Stress}\;{\bar{\sigma }}_{\theta \theta }^{aniso}$$		$$\mathrm{Total}\;\mathrm{Stress}\;\sigma_{\theta \theta }$$		$$\mathrm{Load}-\mathrm{Bearing}\;\mathrm{Fraction}\;\psi_{\theta \theta }^{aniso}$$	
		$$\Delta_{max}$$	$${\hat{R}}^{2}$$	$$\Delta_{max}$$	$${\hat{R}}^{2}$$	$$\Delta_{max}$$	$${\hat{R}}^{2}$$	$$\Delta_{max}$$	$${\hat{R}}^{2}$$
	Set	[kPa]	[-]	[kPa]	[-]	[kPa]	[-]	[-]	[-]
hg-sets	1 (F49)	2.42	0.88	30.44	0.81	44.65	0.58	−0.033	0.87
	2 (M60)	1.42	1.0	2.46	1.0	12.80	0.99	−0.015	1.0
	3 (M61b)	0.14	1.0	75.82	0	90.23	0	−0.015	1.0
	4 (M66)	−0.44	1.0	29.99	0.46	42.44	0.18	−0.016	1.0
	5 (M70a)	2.56	0.86	15.13	0.98	28.05	0.94	−0.03	0.85
	6 (M70b)	−3.93	0.99	50.54	0	62.39	0	0.025	1.0
	median first quartile third quartile	0.78 −0.30 2.17	1.0 0.91 1.0	30.22 18.85 45.52	0.64 0.12 0.94	43.54 31.65 57.95	0.38 0.05 0.85	−0.016 −0.026 −0.015	1.0 0.90 1.0
lc-sets	7 (F50)	2.79	0	52.98	0.44	68.43	0	−0.042	0
	8 (F65)	0.52	0	51.51	0.73	64.81	0.52	−0.005	0
	9 (M57)	0.67	0	103.36	0	117.84	0	−0.005	0
	10 (M61a)	0.19	0	80.44	0	95.29	0	−0.007	0
	11 (M67a)	0.18	0	10.92	0.99	22.09	0.95	−0.006	0
	12 (M67b)	0.20	0	9.65	1.0	20.21	0.98	−0.001	0.23
	13 (M71)	0.15	0	2.26	1.0	11.95	0.99	−0.002	0
	14 (M77)	0.42	0	35.68	0.67	49.37	0.11	−0.01	0
	median first quartile third quartile	0.31 0.18 0.56	0 * 0 0	43.59 10.60 59.84	0.70 0.33 0.99	57.09 21.62 75.14	0.32 0 0.96	−0.006 −0.008 −0.004	0 * 0 0
cs-sets	15 (F63)	3.28	0.99	−2.77	1.0	7.91	1.0	−0.025	1.0
	16 (M38)	1.40	1.0	−1.15	1.0	8.27	1.0	−0.013	1.0
	17	3.74	1.0	−0.85	1.0	8.04	0.99	−0.013	1.0
	18	0.99	1.0	−0.54	1.0	8.03	1.0	−0.003	1.0
	19	1.25	1.0	−0.73	1.0	8.02	0.99	−0.004	1.0
	20	1.99	1.0	−0.66	1.0	8.01	1.0	−0.003	1.0
	21	1.49	1.0	−0.78	1.0	8.11	1.0	−0.006	1.0
	median first quartile third quartile	1.48 1.32 2.64	1.0 1.0 1.0	−0.78 * −1 −0.70	1.0 * 1.0 1.0	8.03 * 8.01 8.07	0.32 * 0 0.96	−0.006 −0.01 −0.004	1.0 1.0 1.0

*: indicate significant differences compared to both other sets, *p* < 0.05.

**Table 4 medsci-13-00009-t004:** Maximum difference ($\Delta_{max}$) and coefficient of determination (${\hat{R}}^{2}$) of predicted and in silico axial measures for each parameter set. For the isotropic stress component, a coefficient of determination cannot be stated because the in silico stress remains constant with respect to blood pressure.

	AXIAL DIRECTION								
		$$\mathrm{Isotropic}\;\mathrm{Stress}\;{\bar{\sigma }}_{zz}^{iso}$$		$$\mathrm{Anisotropic}\;\mathrm{Stress}\;{\bar{\sigma }}_{zz}^{aniso}$$		$$\mathrm{Total}\;\mathrm{Stress}\;\sigma_{zz}$$		$$\mathrm{Load}-\mathrm{Bearing}\;\mathrm{Fraction}\;\psi_{zz}^{aniso}$$	
		$$\Delta_{max}$$	$${\hat{R}}^{2}$$	$$\Delta_{max}$$	$${\hat{R}}^{2}$$	$$\Delta_{max}$$	$${\hat{R}}^{2}$$	$$\Delta_{max}$$	$${\hat{R}}^{2}$$
	Set	[kPa]	[-]	[kPa]	[-]	[kPa]	[-]	[-]	[-]
hg-sets	1 (F49)	6.89	-	5.41	0.96	24.09	0.66	0.144	0.2
	2 (M60)	4.66	-	−6.12	0.99	7.47	1.0	0.083	0.95
	3 (M61b)	7.22	-	29.51	0	51.00	0	0.129	0.94
	4 (M66)	−5.91	-	7.42	0.86	14.37	0.11	0.017	1.0
	5 (M70a)	6.86	-	−1.93	1.0	15.79	0.96	0.121	0.51
	6 (M70b)	4.44	-	17.67	0.28	37.89	0	0.08	1.0
	median first quartile third quartile	5.76 4.49 6.88	- - -	6.42 −0.09 15.11	0.91 0.42 0.98	19.94 14.73 34.44	0.38 0.03 0.88	0.10 0.08 0.13	0.94 0.62 0.99
lc-sets	7 (F50)	10.34	-	15.75	0.80	38.75	0.06	0.198	0
	8 (F65)	11.88	-	15.49	0.90	40.15	0.59	0.189	0
	9 (M57)	14.66	-	40.35	0.39	68.82	0	0.181	0
	10 (M61a)	12.75	-	31.10	0	58.51	0	0.256	0
	11 (M67a)	11.41	-	−2.81	1.0	19.59	0.97	0.255	0
	12 (M67b)	8.77	-	−3.83	1.0	15.30	0.99	0.149	0
	13 (M71)	8.89	-	−6.90	0.99	11.53	1.0	0.177	0
	14 (M77)	12.01	-	9.34	0.90	34.63	0	0.299	0
	median first quartile third quartile	11.64 * 9.98 12.20	- - -	12.42 −3.06 19.59	0.90 0.70 0.99	36.69 8.52 44.74	0.32 0 0.98	0.19 0.18 0.26	0 * 0 0
cs-sets	15 (F63)	−4.02	-	−8.41	0.99	−5.06	1.0	−0.004	1.0
	16 (M38)	−1.52	-	−7.16	0.99	−0.81	1.0	0.015	1.0
	17	10.04	-	−5.16	0.97	9.80	1.0	0.066	0.96
	18	−7.72	-	−5.38	0.97	−5.64	1.0	0.020	0.99
	19	−8.96	-	−5.94	0.93	−7.91	1.0	0.025	0.96
	20	−2.08	-	−5.85	0.97	−1.44	1.0	0.023	0.98
	21	2.54	-	−5.16	0.99	4.67	1.0	0.041	1.0
	median first quartile third quartile	−2.08 −5.87 0.51	- - -	−5.85 −6.55 −5.27	0.97 0.97 0.99	−1.44 * −5.35 1.93	1.0 * 1.0 1.0	0.023 0.018 0.033	0.99 0.97 1.0

*: indicate significant differences compared to both other sets, *p* < 0.05.

**Table 5 medsci-13-00009-t005:** Maximum difference ($\Delta_{max}$) and coefficient of determination (${\hat{R}}^{2}$) of predicted and in silico reaction stress for each parameter set.

		Reaction Stress *p*	
		$$\Delta_{max}$$	$${\hat{R}}^{2}$$
	Set	[kPa]	[-]
hg-sets	1 (F49)	−11.78	0
	2 (M60)	−8.93	0
	3 (M61b)	−14.27	0
	4 (M66)	−12.87	0
	5 (M70a)	−10.36	0
	6 (M70b)	−15.78	0
	median first quartile third quartile	−12.33 −13.92 −10.72	0 0 0
lc-sets	7 (F50)	−12.66	0
	8 (F65)	−12.79	0
	9 (M57)	−13.81	0
	10 (M61a)	−14.67	0
	11 (M67a)	−10.99	0
	12 (M67b)	−10.36	0
	13 (M71)	−9.54	0
	14 (M77)	−13.28	0
	median first quartile third quartile	−12.72 −13.41 −10.83	0 0 0
cs-sets	15 (F63)	−7.39	0
	16 (M38)	−8.03	0
	17	−4.88	0
	18	−7.46	0.95
	19	−7.02	0.51
	20	−6.53	0.83
	21	−7.13	0.84
	median first quartile third quartile	−7.02 −7.13 −6.52	0.21 0 0.84

## Data Availability

The raw data supporting the conclusions of this article will be made available by the authors on request.

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
