# Peer review of "Evaluating the Stress State and the Load-Bearing Fraction as Predicted by an In Vivo Parameter Identification Method for the Abdominal Aorta"

_medsci, 2025, doi:10.3390/medsci13010009_

Round 1
Reviewer 1 Report
Comments and Suggestions for Authors
Dear Authors,
1. Although there is some explanation about the difference between "cs-sets" and "clinical data sets," more detailed information is needed regarding how exactly these two sets differ and how the "cs-sets" are designed. Specifically, it is not entirely clear which parameters are constrained in the "cs-sets," particularly since they do not consider factors such as "sex" and "age," which are included in the clinical data sets.
2. The impact of the "load-bearing fraction" on the mechanical properties of the vessel wall is discussed, but how this parameter is specifically measured and how it functions is somewhat unclear. It would be helpful to provide more detailed explanations, particularly regarding how variations in the "load-bearing fraction" influence pathological conditions or changes in the vessel wall.
3. There should be a more detailed explanation of how collagen contributes to load-bearing in the vessel wall, including its mechanism and role. Additionally, questions may arise about how the load-bearing fraction attributed to collagen is predicted, as well as the methods and accuracy of the calculations involved.
4. The article mentions uncertainties related to parameter identification in the HGO model (Helmholtz-Gregory-Osmolovskyi), but it does not provide specific details on the problems involved or which parameters are particularly uncertain. This lack of detail may raise concerns regarding the precision and reliability of the model's predictions.
5. The paper refers to "experimental validation" and "datasets," but further clarification on the actual testing methods or procedures would help readers understand how to assess and apply this data effectively.
Reviewer 2 Report
Comments and Suggestions for Authors
This study is highly interesting and addresses a significant topic in arterial mechanics, which holds both scientific importance and potential clinical applications. The introduction provides a good overview of the problem, and the results are clearly and transparently presented.
However, it should be noted that the paper lacks a brief paragraph discussing the study's limitations. A significant limitation is the absence of a more comprehensive reference to the mechanics of a full arterial model. The authors should consider the influence of vessel diversity, particularly the fact that smaller vessels exhibit a stronger interaction with the vascular endothelium. These interactions vary depending on the mechanism of contraction initiation. In the primary contraction models — receptor activation (e.g., α1-adrenergic receptors using phenylephrine), G protein activation (e.g., mastoparan-7), or calcium channel activation (e.g., using Bay K 8544) — mechanical effects can differ significantly. A brief discussion of these discrepancies between in silico studies and actual physiological experiments would help readers better understand the problem and the limitations of applying the method to different types of vessels.
In summary, the paper represents a valuable step toward understanding arterial mechanics and its potential application in medical practice. However, it would benefit from the addition of a discussion of these limitations to better contextualize the results and their implications.
Round 2
Reviewer 2 Report
Comments and Suggestions for Authors
The authors have explained in the text the fragments concerning the idea of the study being performed. I believe that the manuscript in its current form can be considered for publication